# An Extensive Study of an Eco-Friendly Fireproofing Process of Lignocellulosic *Miscanthus* × *giganteus* Particles and Their Application in Flame-Retardant Panels

**DOI:** 10.3390/polym17020241

**Published:** 2025-01-19

**Authors:** Yasmina Khalaf, Rodolphe Sonnier, Nicolas Brosse, Roland El Hage

**Affiliations:** 1Laboratory of Physical Chemistry of Materials (LCPM), Campus Fanar, Faculty of Sciences II, Lebanese University, Fanar, Jdeidet P.O. Box 90656, Lebanon; nina_khalaf@hotmail.com; 2Department of Mechanical Wood Technology, Faculty of Forest Industry, University of Forestry, 1756 Sofia, Bulgaria; 3Polymers Composites and Hybrids (PCH), IMT Mines Ales, 6 Avenue de Clavières, 30100 Ales, France; rodolphe.sonnier@mines-ales.fr; 4LERMAB, University of Lorraine, INRAe, GP4W, 54500 Nancy, France; nicolas.brosse@univ-lorraine.fr

**Keywords:** lignocellulosic material, miscanthus, flame retardant, phytic acid, steam explosion, binderless particle panels

## Abstract

Increasing the flame retardancy of lignocellulosic materials such as *Miscanthus* × *giganteus* can effectively enable their wide use. This study examines the fireproofing process of Miscanthus particles using an eco-friendly process by grafting phytic acid and urea in aqueous solution. Miscanthus particles underwent a steam explosion step before being grafted. Fireproof binderless particle panels were manufactured from miscanthus particles with or without adding olive pomace by hot-pressing. The effect of the steam explosion and/or the flame-retardant treatment on the morphology, chemical composition and thermal stability of the particles, as well as the thermal stability of the panels, was investigated. The results showed that water impregnation followed by a steam explosion at 210 °C for 8 min resulted in particles that were rich in lignin and more homogeneous in size (length and width). Fireproof particles were produced with relatively low P and N contents. The flame retardancy of the binderless particle panels was significantly improved when using miscanthus particles treated with phytic acid and urea, as shown by a reduced heat release (HRR) and an increased time-to-ignition. However, the presence of olive pomace significantly decreased the flame retardancy of the panels. Binderless particle panels prepared from grafted miscanthus particles showed the best fire properties and are considered fireproof.

## 1. Introduction

In a world where billions of tons of agricultural residues and wastes are generated and accumulated each year, causing major environmental issues, it is mandatory and advisable to reduce these quantities by finding them novel applications for a safe and green future as their sustainability is guaranteed. Being low cost, widely available, eco-friendly and biodegradable, research in lignocellulosic materials including agricultural residues has grown rapidly.

Flame retardancy is an important and interesting property required in lignocellulosic materials, enabling their wide use. To improve this property, using a flame retardant to modify their chemical structure can be considered [1,2,3]. Among the flame retardants used, phosphorous ones, which are non-halogenated, are increasingly used to improve the fire retardancy performance of lignocellulosic materials. They are considered as effective alternatives to halogenated flame retardants, which release toxic gases during burning [4,5,6]. Among the various phosphorous flame retardants, phytic acid, also called inositol hexaphosphoric acid, is a bio-based, abundant, renewable, high in phosphorus content and environmentally friendly compound [7,8]. In the last decade, it has been used for flame retardancy. Phytic acid was used as a coating treatment in addition to chitosan to improve the fire performance of cotton fabrics [9]. It was also successfully used as a flame retardant for wood [10]. Recently, Antoun et al. developed an easy and efficient eco-friendly process to enhance the flame retardancy of lignocellulosic materials using phytic acid and urea by the covalent grafting of a phytate moiety in an aqueous solution [11]. This solvent-free process results in non-flammable hemp fibres. The study proves the synergistic effect of phosphorous and nitrogen in improving the fire retardancy of hemp fibres. Moussa et al. previously stated that a synergetic P/N effect improved the thermal properties of modified hemp fibres in an aqueous solution of etidronic acid and urea, hence providing an improvement of the fire retardancy of the hemp fibres [12].

Lignocellulosic agricultural biomass materials are considered good substituents to wood in wood-based products such as particle panels [13]. Particle panels represent more than 50% of the global consumption of wood-based panels. Each year, the demand for particleboards grows by about 2–5%. To meet this growing demand without harming the environment by the excessive usage of synthetic adhesives emitting toxic formaldehydes harmful for both humans and the environment, binderless particle panels appear to be good alternatives, since their production does not require the usage of any adhesive. They are considered sustainable and environmentally friendly particleboards [14,15]. In recent years, the production of binderless particle panels and fully lignocellulosic wood panels has attracted more attention. Binderless particle panels using coconut husks, sugarcane bagasse, hemp, spruce and many others have been successfully manufactured [16,17,18,19].

The self-bonding of the natural components present in the lignocellulosic raw material under heat and pressure makes the manufacturing of particle panels without any adhesive possible. The high temperature allows the cross-linking reactions of lignin–lignin and lignin–polysaccharide. Modifying the chemical composition of the lignocellulosic material to obtain higher contents of lignin and carbohydrates is thus beneficial to the self-bonding reactions and therefore to the binderless particle panel production [20,21].

Steam explosion is an efficient, environmentally friendly and industrially scalable pretreatment method to produce binderless panels. It allows the de-structuring of the lignocellulosic materials due to the heat from the steam, followed by explosive decompression to give the lignocellulosic materials the power to stick together under certain conditions, mainly due to lignin depolymerization [22,23].

*Miscanthus* × *giganteus* (M) is a perennial grass that has gained interest in the last decades in panel production since it is a non-edible grass with a rapid growth and high yield that has a chemical composition comparable to that of wood [24]. It has been used alone or mixed with other lignocellulosic materials for the conception of panels with the addition of adhesives like chitosan, poly(butylene adipate-co-terephthalate) (PBAT), casein, gelatine and starch [25,26,27,28]. Steam exploded Miscanthus sinensis particles have been used to produce binderless fibreboards since more than 20 years ago [29,30]. Recently, untreated *Miscanthus* × *giganteus* particles have been used to produce self-binding fibreboards [31]. To the best of our knowledge, Miscanthus has never undergone a fireproofing treatment prior to its usage in particle panel production.

Olive wastes are the leftovers of olive oil production. They are produced yearly in large quantities, making them an environmental hazard due to their richness in phenolic compounds [32,33,34]. Being lignocellulosic waste materials, the different parts of olive wastes (olive pomace (OP), oil-free pomace and olive stones) were recently successfully used in a previous paper as raw materials with other lignocellulosic agricultural waste in the conception of chitosan-based eco-friendly composites [35]. OP has been used as a flame-retardant filler in polymers [36]. The authors found that despite the presence of a high quantity of extractives in OP such as oleic acid, flammability was not affected when being used in moderate amount (10 wt%); thus, they considered OP as a valuable bio-based char source. No research can be found on using any olive wastes in binderless panel production.

The main objective of this work is the implementation of the environmentally friendly fireproofing process proposed by Antoun et al. [11] using phytic acid and urea on steam exploded Miscanthus particles and the optimization of the process. The thermal decomposition of treated particles was studied by pyrolysis combustion flow calorimeter. The phosphorous content was determined by X-ray fluorescence. Their chemical composition was also determined. Treated and non-treated steam exploded particles were used with or without olive pomace in the conception of binderless particle panels to deeply compare their fire behaviour by cone calorimeter. The steam explosion process has been initially optimized. The length and width of the exploded particles as well as their chemical composition have been examined.

## 2. Materials and Methods

### 2.1. Materials

*Miscanthus* × *giganteus* (M) were supplied by EARL Ar Gorzenn (Ferme de la Roselière, 29790 Pont-Croix, France). Olive pomace (OP) was provided by “Ghaoui-Ghaoui” olive oil mill (Darbechtar, Lebanon). This OP was processed in the open air for 7 days before being utilized. Sodium hydroxide (NaOH) having a purity ≥ 99.2% purchased from VWR International (91958 Les Ulis, France) was used as received for particle impregnation. Sulfuric acid (H_2_SO_4_) having a purity of 95% and an average molecular weight of 98.08 g·mol^−1^ was also used as received from VWR Chemicals (91958 Les Ulis, France) for particle impregnation. Urea (CH_4_N_2_O) having a purity ≥ 99.3% purchased from Alfa Aesar chemicals (Haverhill, MA, USA) and phytic acid (C_6_H_18_O_24_P_6_) 50% (*w*/*w*) solution in H_2_O having an average molecular weight of 660.04 g·mol^−1^ purchased from Sigma Aldrich (St. Louis, MO, USA) were used for the fireproofing process.

### 2.2. Olive Pomace Preparation

OP of 7 wt% relative humidity was milled using a Moulinex^®^ coffee grinder (SEB group, based in Ecully, France), manually sieved using a 2 mm pore size sieve to eliminate olive stones and then sieved using a MATEST electromagnetic sieving machine (24048 Treviolo (BG), Italy) to collect the particles retained by the filters, having a porosity of 1.18 mm, to use in panel preparation.

### 2.3. Steam Explosion of Miscanthus Giganteus Particles

The M particles used in this study underwent an impregnation step prior to the steam explosion (SE) using a steam reactor. The particles were cooked using steam in a pressure resistant reactor. The explosion was done by a sudden drop in pressure. The exploded particles were released into the discharge tank. Different impregnation solutions and SE conditions were tested to study their impact on the resulting steam exploded M particles. The different impregnation and SE conditions used are presented in Table 1.

The obtained exploded miscanthus were then washed by filtration through a 40-micrometer sieve and spread out and dried at room temperature. Figure 1 shows the resulting exploded M particles obtained from the six different experimental conditions.

### 2.4. Fireproofing Process of Miscanthus Particles by Phosphorylation

The exploded miscanthus particles of the type Mse-N8-2 (Mse) were treated with an aqueous solution of phytic acid and urea during the fireproofing process. The Mse particles were phosphorylated using various experimental conditions (Table 2). Mse particles were first sprayed with the aqueous solution of phytic acid and urea with different concentrations, as shown in Table 2. Then, the sprayed Mse particles were dried in an oven at a temperature of 60 °C for 15 h prior to a grafting step which was done by cooking the dried Mse at a temperature of 150 °C for a specific time (1 or 2 h, as shown in Table 2). At the end of treatment, the grafted Mse so-called Mse-g particles were washed 3 times abundantly with distilled water then vacuum filtered before being dried in an oven overnight at 60 °C prior to use or analysis.

### 2.5. Binderless Panels Manufacturing

Four formulations of binderless particle panels of 200 × 200 × 5 mm were manufactured with different components as summarized in Table 3. Three panels were prepared of each formulation.

Then, 150 g of particles were weighed (150 g of Mse or Mseg or 90 g of Mse or Mseg together manually mixed with 60 g of OP) and then put into a mould of 20 × 20 mm. The mixture of particles was then pressed using a hot-press at 220 °C. Pressing was carried out by applying a load of 12 MPa for 5 min followed by a one-minute discharge (breathing) at 0 MPa and a second load of 12 MPa for 2 min. After pressing, the obtained panels (Figure 2) were left to cool at room temperature before characterization.

### 2.6. Characterization Techniques

#### 2.6.1. Chemical Composition

The chemical composition in extracts, monomer sugars and lignin of the lignocellulosic steam exploded miscanthus particles with different conditions has been estimated.

Dichloromethane (DCM) extract contents were measured on 2 g of each miscanthus sample under reflux for 6 h using a Soxhlet apparatus with dichloromethane. They were calculated according to Equation (1):(1)$$DCM\;extracts\;content\;\left(\%\right)=100\cdot \frac{m_{e}}{m_{i}}$$
with *m_e_* being the mass in g of dry extracts-free particles and *m_i_* being the mass in g of the initial native dry particles.

The carbohydrate and Klason lignin contents were measured in extract-free particles.

The Klason lignin was measured according to the laboratory analytical procedure (LAP) provided by the National Renewable Energy Laboratory (NREL). First, 1.5 mL of 72% sulfuric acid was used to hydrolyse 0.175 g of the sample in a water bath at 30 °C for 1 h. Then the sample was diluted with 42 mL of deionized water and autoclaved for 1 h at 120 °C before being filtered and dried for 2 h at 105 °C. The Klason lignin content was calculated according to Equation (2):(2)$$Klason\;Lignin\;\left(\%\right)=100\cdot \frac{m_{L}\left(g\right)}{m_{i}\left(g\right)}$$
where *m_L_* is the mass of lignin recovered and *m_i_* is the initial dry mass of the extract-free particles.

The monomer sugar content in the liquid fraction obtained after filtrating and washing the sample was quantified using high performance anion-exchange chromatography with pulsed amperometry detection HPAEC 6 PAD, ICS-3000 Dionex (Thermo Fisher Scientific, Waltham, MA, USA).

#### 2.6.2. Elementary Analysis

The elemental analysis of carbon, nitrogen, oxygen and hydrogen was done on the steam exploded miscanthus particles before and after fireproofing treatment using a Thermo Finnigan Flash EA 112 Series (Thermo Fisher Scientific, Waltham, MA, USA). The combustion of the samples to be analysed (1.5 mg) was carried out at a high temperature (1000 °C) in the presence of tungstic anhydride under an oxidizing atmosphere for 15 s. This decomposition produces many gaseous products (CO_2_, H_2_O, SO_2_ and NO_x_, which is reduced to N_2_ in the presence of copper), which were analysed by gas chromatography. The results were recorded and analysed by the “Eager 300” software (version 2.3), which directly calculates the mass percentage of each element present in the compound.

#### 2.6.3. X-Ray Fluorescence (XRF)

The phosphorous content of the treated particles was identified using the ED-XRF QUANT’X (ThermoFischer, Waltham, MA, USA) non-destructive device, which is based on counting the fluorescent X-rays emitted by the sample after being excited by a primary X-ray source. The particles were irradiated with X-rays and each spectrum was collected for 8 min. The mass percentage of phosphorous was determined by a simple calculation of the data collected, as the intensity of the phosphorous peak is proportional to its concentration. Each sample was analysed 3 times to ensure the reproducibility of the measurements.

#### 2.6.4. Length and Width Measurements

Length and width measurements were done on different steam exploded miscanthus particles and on raw miscanthus particles as received. Particles of each kind were scanned in 5 different batches on A4 paper using an office scanner and then 100 measurements of length and width for each sample were done using a ruler.

#### 2.6.5. Pyrolysis-Combustion Flow Calorimetry (PCFC)

A combustion microcalorimeter (Fire Testing Technology Ltd., East Grinstead, UK) was used to study the fire behaviour of grafted Miscanthus particle samples at the micrometric scale (2–4 mg). The samples were pyrolyzed under a flow of nitrogen (100 mL/min) with a temperature rise rate of 1 °C/s from 90 to 750 °C (anaerobic pyrolysis). The pyrolysis gases were transported to a combustion chamber in the presence of a flow of N_2_/O_2_ (80/20), allowing all gases to be fully oxidized. Each sample was tested twice to ensure the reproducibility of our measurements. The peak of heat release (pHRR), temperature at pHRR (TpHRR), total heat release (THR), heat of complete combustion (Δh) and final residue rate (%) were determined.

#### 2.6.6. Preliminary Fire Test on Particle Panels

A non-standardized fire test was carried out to evaluate the flammability of particle panels in a simple and rapid manner. The panels were put vertically on an aluminium support and ignition tested by applying a flame at the top of each sample for 30 s using a gas torch flame gun. Three behaviours are thus distinguished: flame propagation without the formation of a stable residue (flame spreading), flame propagation with the formation of a stable residue leading to self-extinguishing (self-extinguishing) and an absence of flame catching (non-flammability).

#### 2.6.7. Cone Calorimetry

A cone calorimeter was used to evaluate the fire behaviour of the particle panels. This technique consisted of heating, in the presence of air (air flow rate 24 L/s), samples with a surface area of 100 × 100 mm^2^. The samples decomposed and released combustible gases which ignite in the presence of a spark. The panels were tested under a heat flux of 35 kW/m^2^, corresponding to a developing fire. The distance between the radiative source and the sample was 25 mm. The peak heat rate (pHRR), time to ignition (TTI), total heat released (THR), effective heat of combustion (EHC), final residue rate and total smoke released (TSR) were determined.

## 3. Results and Discussion

### 3.1. Steam Explosion Effect on Miscanthus Particles

The effects of different steam explosion pretreatment conditions, including the impregnation solution nature (acid, base or water), steam temperature and pretreatment time (residence time), on the chemical composition, morphology and size of the obtained Miscanthus exploded particles were evaluated. Table 4 summarizes the chemical composition of the various obtained steam exploded Miscanthus particles and the raw M. As a lignocellulosic material, M contains cellulose, hemicelluloses, lignin and DCM extracts in different proportions. Its composition was comparable to that obtained in the literature [37]. According to the literature, miscanthus also contains traces of inorganic components in the form of heavy metals like lead, iron and copper when grown in contaminated soils [38].

It was found that the impregnation of M particles in NaOH prior to SE treatment allowed a recovery of the highest content of cellulose (about 62%) and the least content of lignin (about 5%) in the obtained Mse-B-1. Contrariwise, the acidic impregnation of M particles (Mse-A-1) resulted in increasing the lignin and cellulose content from about 27% to 30% and from 40 to 53%, respectively, with a decrease of hemicellulose content from 21% to 4%. These results are confirmed by a study where the authors studied the influence of steam explosion treatment combined with sulfuric acid pre-impregnation on miscanthus, poplar and wheat straw [39]. It was also found that the impregnation in water in all the experimental conditions tested led to an increase of cellulose content and a decrease of hemicellulose content. It was also clear that when the pretreatment temperature increased from 190 to 210 °C, the lignin content and the cellulose content increased while the hemicellulose content decreased. The pretreatment residency time also affected the chemical composition of exploded particles, as its increase from 4 min to 8 min led to an increase of lignin content and a decrease in hemicellulose content, while the cellulose content seemed to remain almost the same. These results are in accordance with those obtained by Bhatia et al., who evaluated the composition of miscanthus particles before and after steam explosion and found that the cellulose content increased from 37% to 40–45% and Klason lignin increased from 21–23% to 24–27% after SE pretreatment [40]. Concerning the DCM extract content, it was found that it is very variable with the treatment type as it changed from 0.49% for Mse-N4-2 in neutral water impregnation to 13.13% for Mse-A-1 particles that had undergone acidic pre-impregnation. According to the literature, hemicellulose fraction is easily hydrolysed during the steam explosion treatment while cellulose degradation is limited and requires high treatment severity, which explains the increase in cellulose content after the treatment. The depolymerisation of lignin due to the high reactivity of its hydroxyl groups or repolymerisation by a C-C bond formation may be also induced [12,23,41,42]. These previous discoveries may provide a comprehensive explanation for all the results observed in this study.

For particle shapes, a dough-like miscanthus was obtained after the acidic and basic impregnation of M particles, as shown previously in Figure 1 (Mse-B4-1 and Mse-A4-1) of the different steam exploded miscanthus particles obtained. The water impregnation leads to distinct miscanthus particles of different shape and colour (see Figure 1 (Mse-N4-1, Mse-N4-2, Mse-N8-1, and Mse-N8-2)). Mse-B4-1 is a white colour, which is caused by the NaOH impregnation, which leads to the low content of surface lignin (about 5%) responsible for the colour. Contrariwise, all the other steam exploded miscanthus obtained are a brown colour, caused by the high content of surface lignin (between 20 and 36%).

Figure 3A,B show, respectively, the box plots of the length and width of particles obtained after water impregnation and steam explosion. These box plots highlight a clear statistical observation, taking into consideration the value distributions of length and width. The results show a reduction of the length and width of M particles after SE treatment in comparison with those of the untreated M particles. The median value of particle length decreased from about 15 mm for raw M to 7 mm for Mse-N8-2 and the median value of particle width decreased from about 2 mm for raw M to about 0.4 mm for Mse-N8-2.

Since a high lignin content is better for the preparation of binderless particle panels [20,43] and since a more homogeneous particle size is needed, Mse-N8-2 particles, which were the result of the water impregnation of M particles followed by a steam explosion treatment at 210 °C for 8 min and showed the highest content of lignin (36%), were afterwards used in the fireproofing process and in the conception of the binderless particle panels.

### 3.2. Fireproofing Effect on Miscanthus Particles

The effect of the different phosphorylation conditions on steam exploded miscanthus particles has been evaluated. The PCFC data are gathered in Table 5. The nitrogen and phosphorous contents grafted onto the exploded miscanthus particles are mentioned in Table 2. Comparing the seven batches, phytic acid was the source of grafted phosphorous, with 1 wt% phosphorous for Mse-g6 compared to approximately 0 wt% phosphorous for the untreated Mse. The phosphorous content (grafted P on the Mse particles) increased with the concentration of phytic acid used in the impregnation solution. Likewise, urea was the source of nitrogen, with an approximately constant weight percentage of grafted nitrogen on the three batches of particles grafted during the same cooking duration (1 or 2 h) with the same rate of nitrogen (10%) (see Table 2). These results agree with the work of Antoun et al., who stated that the grafted phosphorous content (P%) increased with the phytic acid concentration [11]. It was also clear that increasing the cooking duration resulted in an increase of the grafted phosphorous and nitrogen content. For example, by comparing the P and N content for Mse-g1 and Mse-g4 having the same phytic acid and urea content but different cooking durations, as shown in Table 2, the N and P content increased by 17% and 31%, respectively, when the cooking duration increased from 1h to 2 h. Comparing these results with the literature for the same phytic acid and urea used, it can be stated that the phosphorous-grafted content was lower for the steam exploded miscanthus than for steam exploded hemp shives and similar to the phosphorous content obtained for spruce [19]. Many studies state that the reactions of phosphorous molecules, in the presence of urea, with polysaccharides like cellulose produce various chemical groups, such as phytate ester, carbamylated glucose, ammonium phosphate and phytic pyrophosphate, and a strong hydrogen bond between the amide group of carbamoylated glucose and phosphate groups [11,44,45,46,47].

Figure 4 shows the HRR curves for the seven batches of particles tested in PCFC. HRR curves as a function of temperature (Figure 4) show that the pHRR intensity decreases sharply with increasing phosphorous content from 178 W/g for Mse (non-grafted) to 107 W/g for Mse-g6 with 1 wt% phosphorous. The pHRR temperature also decreases from 381 to 288 °C. The same trend is seen for the total heat release THR, which decreases by 49% upon reaching the threshold of 6 kJ/g for Mse-g6 grafted with 1 wt% of phosphorous compared to 12.4 kJ/g for untreated Mse (Table 5). Conversely, the greater the percentage of phosphorous grafted onto the miscanthus particles, the greater the percentage of residue increases, reaching 28 wt% for Mse-g6 grafted with 1 wt% phosphorous. The decrease in pHRR and THR values observed can be explained by the increase in particle charring due to phosphorous grafting. The increase of char content, knowing that it is rich in carbon, leads to the production of less combustible (carbon-rich) gases. It was seen that grafting of urea/PA (phosphorylation) leads to a reduction in thermal stability, pHRR, THR and ΔH and to an increase in char content. These observations agree with those obtained for hemp fibres modified with urea/PA [11] and phosphorylated flax with dimethyl(methacryloxy)methyl phosphonate (MAPC1) or dimethyl vinyl phosphonate (MVP) flame retardants [6]. However, it can be noted that, for the same phosphorous content, the results obtained for miscanthus particles in term of pHRR, THR and ΔH are higher than those obtained for hemp fibres with a similar residue rate [11]. The flammability of miscanthus is therefore mainly influenced by the phosphorous level, which agrees with the study which found that the level of fireproofing often depends on the phosphorous level [48].

According to the obtained results, an aqueous solution containing 20 wt% phytic acid and 10 wt% urea and a sufficient cooking step of 2 h are necessary to increase the grafting of the nitrogen and phosphorous on the miscanthus particles and to obtain fireproof particles.

### 3.3. Flame Retardancy of Binderless Particle Panels

The prepared binderless particle panels of this study had average densities ranging from 697 to 717 kg/m^3^. The effect of using miscanthus particles previously treated with phytic acid and urea (phosphorous grafted) on the fire behaviour of the produced particle panels was evaluated by a non-standardized flammability test and cone calorimeter test. Figure 5 presents the binderless particle panels before and after applying the flame with a gas torch flame gun. It was first observed that the panels containing treated miscanthus particles with phytic acid exhibited a slightly darker colour, caused by reactions between nitrogen/phosphorous flame retardants and the miscanthus components [49]. The panels clearly showed three different behaviours. Panels containing exploded miscanthus particles untreated with phytic acid and urea with or without OP (Mse and MseOP panels) were easily flammable and had lost an important quantity of matter at the end of the test. However, particleboards containing the phosphorous-grafted miscanthus and olive pomace (MsegOP) were self-extinguishing, as they ignited but extinguished rapidly. Only the Mseg panels containing phosphorous-grafted miscanthus particles were fireproof (non-flammable) as they did not even capture the fire and the flame did not spread, showing thus an excellent fire behaviour.

Table 6 presents the main data obtained by cone calorimeter, including the peak heat rate (pHRR), time to ignition (TTI), total heat released (THR), effective heat of combustion (EHC) and final residue rate. Figure 6 shows the evolution of the HRR curves at an irradiance (heat flux) of 35 kW/m^2^ of the different particle panels. The heating of the plate leads to the appearance of fine gas bubbles. Ignition occurs when the resulting volatile gases migrate freely to the surface and react with oxygen in the air. The time to ignition (TTI) is determined visually (by the appearance of a flame) and corresponds to the point from which an increase in heat flux was recorded in the HRR curves. Resistance to ignition is expressed as a function of the ignition delay (or time to ignition) when the particle panel is exposed to the spark [50]. Table 6 and Figure 6 obviously show that the time to ignition (TTI) of the panels varies depending on their composition. The TTI of Mse particle panels containing 100% non-grafted miscanthus particles at 35 kW/m^2^ is 76 s, whereas the TTI reaches 192 s for Mseg, the particle panels containing phosphorous-grafted miscanthus particles. It is noted that the ignition is accelerated in the presence of the OP, probably due to the presence of the highly flammable oil it contains [36]. Particle panels containing OP have the lowest TTI values of 32.5 s for MseOP panels. A small increase of 18.5% in TTI value is detected for MsegOP, which reaches 38.5 s, caused by the presence of the phosphorous-grafted miscanthus particles.

The Mse, MseOP and MsegOP panels show double-peaked HRR curves (Figure 6). The first peak occurs just after ignition. It is followed by a reduction in the heat flux detected in the HRR curves of Mse and MsegOP panels due to carbonization on the surface forming a “barrier layer” which slows down heat transfer to the bulk and therefore the process of thermal decomposition and formation of volatile substances. The second peak appears as the decomposition continues, through the thickness, to the bottom face of the panels. In the case of the MseOP panels, no reduction of the heat release (HRR) can be seen in the central part of the HRR curve and the HRR continues to increase between the first and the second peak. Finally, the drop in HRR is caused by the extinction of the flame. A different behaviour of the HRR curves is observed for the Mseg panels containing 100% phosphorous-grafted miscanthus particles, as only one peak of HRR is detected and the second peak is eliminated, accompanied with a visual observation of a ghost-like dancing flame hovering lightly above the surface without ever “touching” the plate. This ghost flame highlights that the conditions for a stable flame were not reached (the sample was very close to its self-extinguishment threshold) (see Appendix A). The pHRR is very low and the flaming period is very short, highlighting the excellent flame retardancy of this panel.

Comparing the results obtained for the Mse and Mseg particle panels (Table 6), it can be noted that the pHRR1 decreases by 65% from 162 kW/m^2^ for Mse to 56 kW/m^2^ for Mseg. A strong decrease in EHC (and subsequently in the THR) from 14.1 kJ/g Mse to 1.3 kJ/g for Mseg is observed due to the presence of phosphorous in Mseg, which promotes carbon-rich charring, but also due to the very short flaming period. The phosphorous improved the fire behaviour of the panels by decreasing steadily the peak of heat release rate and the total heat release and shifting the time to ignition and the time to peak of heat release rate remarkably. These results totally agree with the literature stating that the phosphorous level influences the flammability of lignocellulosic materials [5,6,19]. The smoke production (TSR) reaches 250 m^2^/m^2^ for the Mse panels but increases for Mseg to 395 m^2^/m^2^. Indeed, the smoke production rate for these panels is high during the non-flaming period but sharply drops after ignition and up to flame-out (Figure 7). This means that the soot (which is responsible for smoke opacity) can be destroyed in the flame (due to the high temperature). While the non-flaming period is particularly long for these panels, the TSR is the highest for the Mseg panels.

The results obtained for the MseOP panels show an increase in pHRR1 to 183 kW/m^2^, in pHRR2 to 279 kW/m^2^, EHC to 16 kJ/g, THR to 13.7 kJ/g and TSR to 373 m^2^/m^2^ in comparison with the results obtained for the Mse particleboards that do not contain olive pomace, but all these values remain lower (or similar) for MsegOP panels than for Mse. These observations are linked to the presence of extractives in OP in a high percentage (26.28%). According to the literature, olive pomace presents two HRR peaks (86 and 74 kW/m^2^). The first one is related to extractives having a high THR (>30 kJ/g). Extractives contain mainly oleic acids and oils which are known for their high flammability [51].

For the panels containing miscanthus particles treated with phytic acid, the mode-of-action of phosphate groups present in phytic acid promotes the carbonization of cellulose. Indeed, the dehydration of polymer (cellulose) chains by phytic acid leads to C=C double bonds. These double bonds carried by adjacent polymer chains further allow the formation of thermally stable polyaromatic compounds (i.e., char). A large amount of carbon is stored in the char, leading to a strong decrease in EHC (i.e., in the energy contained in the gas fraction and released by gas-phase combustion). Moreover, a protective carbon layer that acts as a physical barrier between the gaseous and condensed phases is formed when the panel is exposed to a heat source. The whole process leads to a drop in the decomposition kinetics and the quantity of pyrolysis gases and their energy, thus leading to the prevention of the ignition or the stopping of the flame propagation [5,19,52].

Photographs of the different particle panels after the cone calorimeter test are shown in Figure 8. The Mse panels (Figure 8A) were not able to retain their original structure and only yielded a residue of 10.5 wt% (Table 6). A brittle behaviour was also seen for the MseOP panels (Figure 8B), with a residue of 14 wt%. On the other hand, the Mseg and MsegOP panels, which contain phosphorous-grafted miscanthus particles, showed a maintained structure and had a more stable residue of 44.7 wt% and 33.2 wt%, respectively (Figure 8C,D). The phosphorous protects the residue from thermo-oxidation and the residue is still black (carbon-rich), as proved in the literature [11].

## 4. Conclusions

This study examined the implementation of a novel fireproofing process using phytic acid and urea on steam exploded Miscanthus particles as well as their usage in flame-retardant binderless particle panels with or without adding olive pomace. The steam explosion process, the fireproofing process and the preparation of panels have been thoroughly analysed. Results show that the water impregnation of the miscanthus particles, followed by a steam explosion at 210 °C for 8 min, leads to particles that are richer in lignin and more homogeneous in size (length and width), which is beneficial for binderless particle panel preparation. The impregnation of the steam exploded particles in an aqueous solution containing 20 wt% phytic acid and 10 wt% urea is necessary to produce fireproof particles with relatively low phosphorous and nitrogen contents (1 wt% P and 1.5 wt% N). In addition, the flame retardancy of the produced panels was significantly improved when using miscanthus particles treated with phytic acid and urea. However, the presence of olive pomace in the panels reduced their flame retardancy. The panels prepared from grafted miscanthus particles showed the best fire properties and are considered fireproof.

This study illustrates that this eco-friendly fireproofing process using phytic acid and urea can be effectively utilized to make Miscanthus particles fire retardant. It also proves the feasibility of their usage in fireproof binderless particle panels. Future research could focus on testing and optimizing the mechanical properties and durability of these panels to meet the international standards for particle panels.

## Figures and Tables

**Figure 1 polymers-17-00241-f001:**
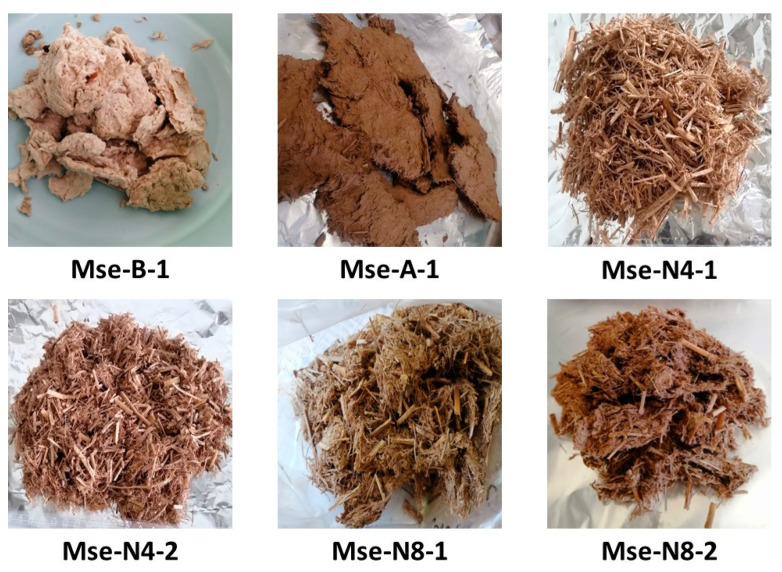
The different steam exploded miscanthus particles obtained.

**Figure 2 polymers-17-00241-f002:**
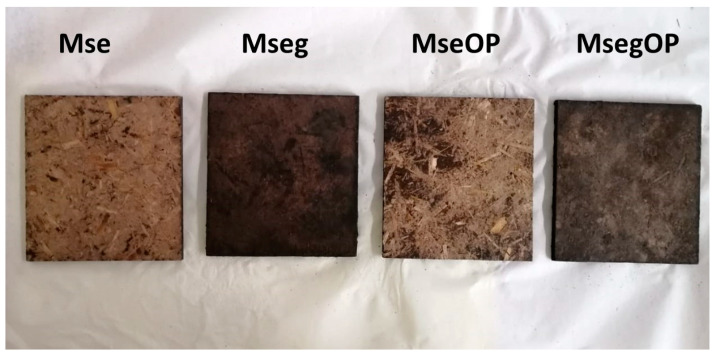
The different panels produced.

**Figure 3 polymers-17-00241-f003:**
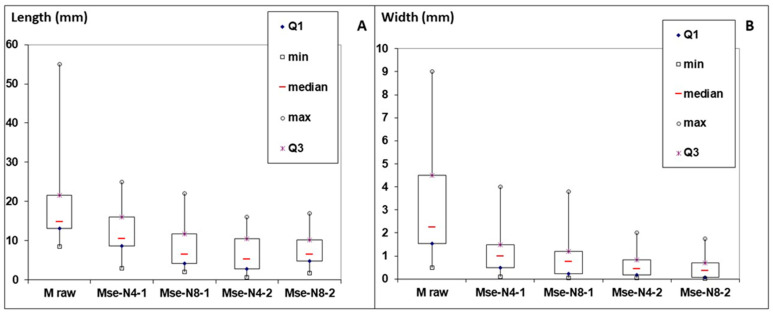
Box plots representing the sizes of raw M and the different M exploded particles: (**A**) length of particles (mm); (**B**) width of particles (mm).

**Figure 4 polymers-17-00241-f004:**
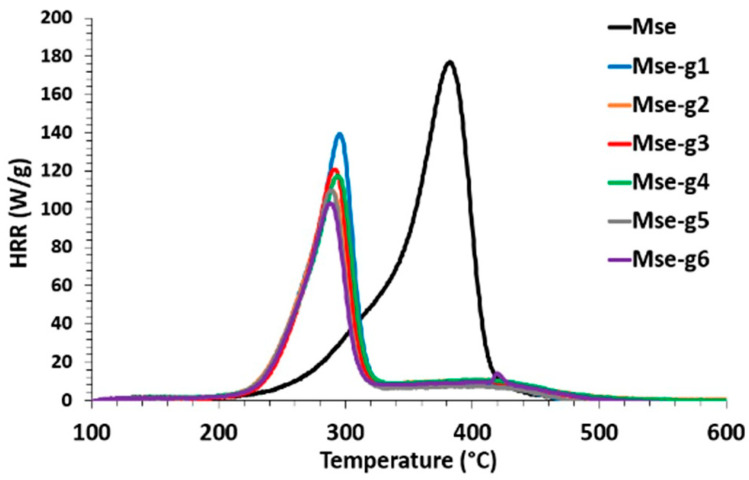
Evolution of the heat release rate (HRR) as a function of temperature for exploded miscanthus particles with and without grafting.

**Figure 5 polymers-17-00241-f005:**
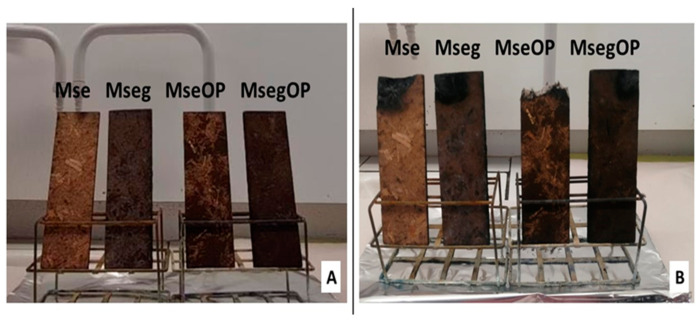
Binderless particle panels (**A**) before the flammability test and (**B**) after the flammability test.

**Figure 6 polymers-17-00241-f006:**
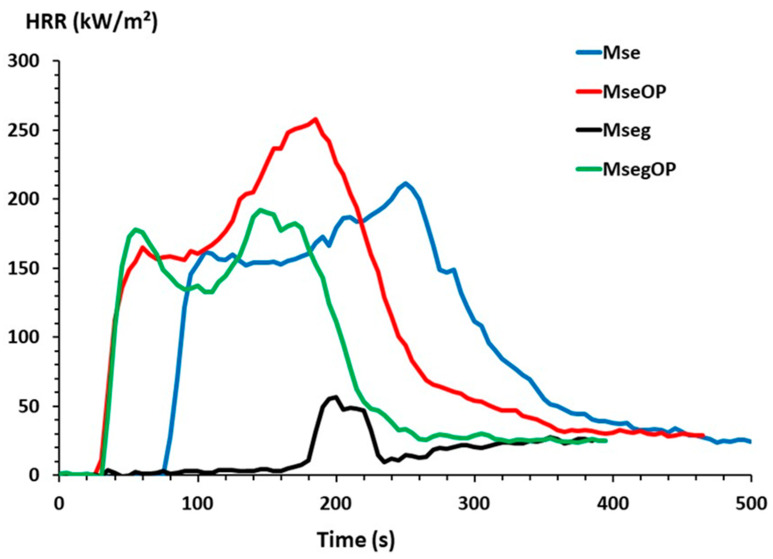
HRR curves obtained by cone calorimeter at an irradiance of 35 kW/m^2^ for all panels.

**Figure 7 polymers-17-00241-f007:**
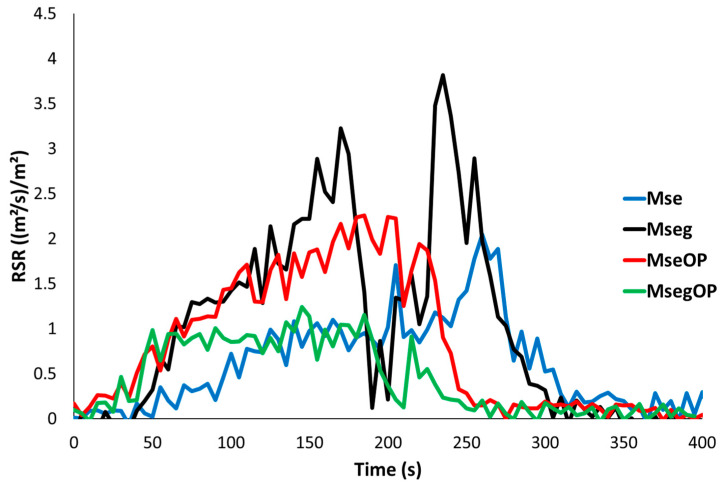
Rate of smoke released (RSR) curves obtained by cone calorimeter at an irradiance of 35 kW/m^2^ for all panels.

**Figure 8 polymers-17-00241-f008:**
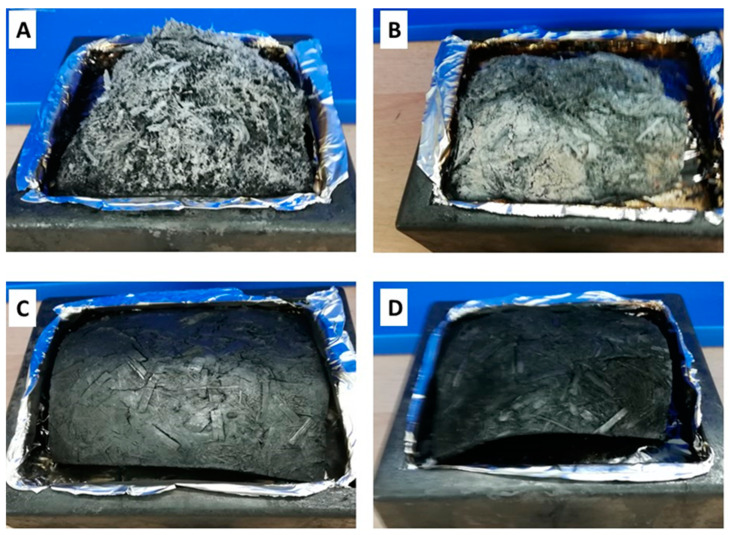
Photographs of the binderless particle panels after the cone calorimeter test ((**A**): Mse; (**B**): MseOP; (**C**): Mseg; and (**D**): MsegOP).

**Table 1 polymers-17-00241-t001:** Impregnation and steam explosion conditions of miscanthus particles.

Sample Name	Type of Impregnation	Steam Temperature (°C)	Pretreatment Time (min)
Mse-B-1	NaOH 8%	190	4
Mse-A-1	H_2_SO_4_ 1%	190	4
Mse-N4-1	H_2_O	190	4
Mse-N4-2	H_2_O	190	8
Mse-N8-1	H_2_O	210	4
Mse-N8-2	H_2_O	210	8

Duration of impregnation is 15 h for all samples.

**Table 2 polymers-17-00241-t002:** Experimental conditions used to study the fireproofing process as well as the nitrogen and phosphorous contents of the resulting miscanthus particles.

Sample Name	Phytic Acid (wt.%)	Urea (wt.%)	Cooking Temperature (°C)	Cooking Duration (h)	Washing 3 Times	N (wt%)	P (wt%)
Mse	0	0	-	-	-	0	0.01
Mse-g1	5	10	150	1	+	1.05	0.49
Mse-g2	10	10	150	1	+	1.07	0.66
Mse-g3	20	10	150	1	+	1.03	0.76
Mse-g4	5	10	150	2	+	1.23	0.64
Mse-g5	10	10	150	2	+	1.39	0.96
Mse-g6	20	10	150	2	+	1.25	0.99

**Table 3 polymers-17-00241-t003:** Designation of the binderless particle panels and their corresponding formulations and densities.

Name of the Formulation	Components Composition	Olive Pomace Ratio (wt%)	Density (kg/m^3^)
Mse	Steam exploded Miscanthus	0	699 ± 33
MseOP	Steam exploded Miscanthus/olive pomace	40	697 ± 43
Mseg	Phosphorous-grafted steam exploded Miscanthus	0	702 ± 53
MsegOP	Phosphorous-grafted steam exploded Miscanthus/olive pomace	40	717 ± 30

**Table 4 polymers-17-00241-t004:** Chemical composition of raw miscanthus particles and miscanthus particles after SE pretreatment.

Sample Name	Cellulose (%)	Hemicellulose (%)	Lignin (%)	DCM Extracts (%)
M	40.07 ± 2.54	21.20 ± 1.40	26.72 ± 0.37	0.95 ± 0.01
Mse-B-1	62.11 ± 9.73	3.09 ± 0.75	5.29 ± 0.31	2.69 ± 0.21
Mse-A-1	53.27 ± 2.41	3.75 ± 1.15	30.5 ± 1.63	13.13 ± 0.18
Mse-N4-1	44.72 ± 1.06	21.19 ± 1.54	20 ± 0.67	6.9 ± 0.34
Mse-N4-2	46.1 ± 2.17	11.5 ± 0.79	22.09 ± 0.33	0.49 ± 0.02
Mse-N8-1	45.06 ± 2.25	15.29 ± 1.65	27.32 ± 0.46	1.75 ± 0.1
Mse-N8-2	47.43 ± 2.75	8.55 ± 1.3	36.19 ± 1.77	5.36 ± 0.42

**Table 5 polymers-17-00241-t005:** Main PCFC data of the different grafted and non-grafted miscanthus particles.

Sample Name	PCFC				
	pHRR (W/g)	TpHRR (°C)	THR (kJ/g)	Residue (%)	∆H (kJ/g)
Mse	178	381	12.4	7.4	13.3
Mse-g1	136	295	7.6	25.8	10.2
Mse-g2	110	290	6.3	25.6	8.4
Mse-g3	125	292	6.7	26	9
Mse-g4	112	290	6	26.3	8
Mse-g5	108	287	6.1	27.8	8.5
Mse-g6	107	288	6	28	8.4

**Table 6 polymers-17-00241-t006:** Main data measured by cone calorimeter for the binderless particle panels.

	Mse	MseOP	Mseg	MsegOP
TTI (s)	76	32.5	192	38.5
pHRR1 (kW/m^2^)	162	183	56	144.5
pHRR2 (kW/m^2^)	231.5	279	-	245
THR (MJ/m^2^)	46.75	50.3	5.8	33.4
THR (KJ/g)	12.6	13.7	0.7	8.6
EHC (kJ/g)	14.1	16	1.3	12.8
Residue (%)	10.5	14	44.7	33.2
TSR (m^2^/m^2^)	250	373	395	260

## Data Availability

All data supporting the findings of this study are provided within the article and the Appendix A.

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
