# Peer review of "An Extensive Study of an Eco-Friendly Fireproofing Process of Lignocellulosic Miscanthus × giganteus Particles and Their Application in Flame-Retardant Panels"

_polymers, 2025, doi:10.3390/polym17020241_

Round 1

Reviewer 1 Report (Previous Reviewer 1)

Comments and Suggestions for Authors

I think that the manuscript has been revised to meet the requirements for publication.

Comments on the Quality of English Language

Need to improve English writing skills.

Author Response

Thank you for your positive feedback and for taking the time to review the manuscript. We truly appreciate your guidance throughout this reviewing process.

Reviewer 2 Report (New Reviewer)

Comments and Suggestions for Authors

The manuscript "polymers-3403403" by Khalaf et al. reported an extensive study of an eco-friendly fireproofing process of lignocellulosic Miscanthus x Giganteus particles and their application in flame retardant panels. The authors have to make major changes. The authors should refer to the following comments to improve their work:

1. Page 7, lines 275 to 295: In the explanation of Table 4, there is no clear explanation of why the NaOH and acid treatments reduce lignin. Instead of comparing the results with other studies, I suggest that you clearly explain the reasons for the changes in lignin, cellulose, hemicellulose, and DCM extract.

2. How did you obtain the particle dimensions (Figure 3)? Take pictures of all samples using a microscope (regular microscope) and add them to the manuscript.

3. What applications do you have in mind for the final product

4. Explain in a schematic the addition of nitrogen and phosphorus content grafted onto the miscanthus particles.

5. Although at the end of the manuscript, the authors have mentioned the investigation of mechanical properties in future research, it is very necessary to add at least one test to this manuscript. I suggest adding the stress-strain curve (tensile test) of the samples. Because it is necessary to see what mechanical properties the samples have and if can they be used in practice. Because this research was conducted for practical purposes and therefore a study of mechanical properties must be added.

6. The language of the manuscript should be checked.

7. Please replace the up-to-date references.

Comments on the Quality of English Language

The language of the manuscript should be checked.

Round 2

Reviewer 2 Report (New Reviewer)

Comments and Suggestions for Authors

Accept in present form

This manuscript is a resubmission of an earlier submission. The following is a list of the peer review reports and author responses from that submission.

Round 1

Reviewer 1 Report

Comments and Suggestions for Authors

I believe that this manuscript lacks innovation and lacks a description of the degradation mechanism of flame retardant panels after combustion and does not characterize the amount of smoke released from flame retardant panels after cone calorimetry testing. 

Comments on the Quality of English Language

The level of writing and presentation of this manuscript needs to be improved.

Reviewer 2 Report

Comments and Suggestions for Authors

Miscanthus as a traditional plant from Asia has specifics in its distribution. Usually, miscanthus is collected in the second or subsequent years. The share of cellulose in it reaches 40-50% and therefore in some countries this plant is grown on an industrial scale. One of the areas of use of miscanthus is its use as fire-protective materials.

In Table 3, it is desirable to provide individual values for olive pomace.

The methodological part is presented in full, i.e. the methods and materials are described in detail. The only remark is the chemical analysis of the raw materials used. Namely, the content of inorganic compounds (metals). It is necessary to pay attention to this work - https://doi.org/10.3390/polym16202915 which indicates metals in miscanthus. And as is known, the presence of metals will dramatically affect the thermal behavior of lignocellulosic materials.

Section 2 is presented at a good level. The presented descriptions, although it is possible to expand, are not necessary.

All the presented results are adequately described, but as I understand it, the obtained results have practical significance. I miss a comparison with other systems based on cellulose and its impurities, for example, is it possible to use reeds or nettles? What influences the observed results? How does the place of harvesting affect the obtained data?

The conclusions and the list of references also meet the requirements of the journal.